# Milk Yields and Milk Fat Composition Promoted by Pantothenate and Thiamine via Stimulating Nutrient Digestion and Fatty Acid Synthesis in Dairy Cows

**DOI:** 10.3390/ani13152526

**Published:** 2023-08-05

**Authors:** Jing Zhang, Yapeng Liu, Lijun Bu, Qiang Liu, Caixia Pei, Gang Guo, Wenjie Huo

**Affiliations:** College of Animal Science, Shanxi Agricultural University, Jinzhong 030801, China; zhangjingdongke@163.com (J.Z.); m13935911570@163.com (Y.L.); m13223543917@163.com (L.B.); caixiapeisxnd@163.com (C.P.); guosteel1984@163.com (G.G.); huohuo-1982@163.com (W.H.)

**Keywords:** coated pantothenate, coated thiamine, milk production, nutrients digestion, fatty acid synthesis

## Abstract

**Simple Summary:**

Coated calcium pantothenate or coated thiamine supplementation potentially promotes milk production by stimulating nutrient digestion and fatty acids synthesis of the mammary gland in dairy cows. Owing to the synergistic effect of pantothenate and thiamine on the regulation of energy metabolism, the combined addition of coated calcium pantothenate and coated thiamine will more effectively promote lactation performance than that achieved by coated calcium pantothenate or coated thiamine alone. Further research is needed to determine the influences of coated calcium pantothenate or/and coated thiamine on the development of the mammary gland.

**Abstract:**

Considering the synergistic effect of pantothenate and thiamine on the regulation of energy metabolism, this study investigated the influences of coated calcium pantothenate (CCP) and coated thiamine (CT) on milk production and composition, nutrients digestion, and expressions of genes involved in fatty acids synthesis in mammary glands. Forty-four multiparous Chinese Holstein cows (2.8 ± 0.19 of parity, 772 ± 12.3 kg of body weight [BW], 65.8 ± 8.6 days in milk [DIM] and 35.3 ± 1.9 kg/d of milk production, mean ± SD) were blocked by parity, BW, DIM, and milk production, and they were allocated into one of four treatments in a 2 × 2 factorial block design. Additional CCP (0 mg/kg [CCP−] or 55 mg/kg dry matter [DM] of calcium pantothenate from CCP [CCP+]) and CT (0 g/kg [CT−] or 5.3 mg/kg DM of thiamine from CT [CT+]) were hand-mixed into the top one-third of total mixed ration. Both CCP and CT additives increased milk production, fat content, true protein, and lactose by promoting nutrient digestibility. The CCP or/and CT supplementation induced the elevation of C11:0, C12:0, C13:0, C14:0, C14:1, C15:0, C15:1, C16:00, C16:1, C24:00, C24:1 fatty acids, saturated fatty acid, and C4-16 fatty acid contents in milk fat; but it decreased C17-22 fatty acid content. Ruminal total VFA content was increased, but pH was decreased by both additives. The ruminal fermentation pattern was altered, and a tendency of acetate formation was implied by the increased acetate-to-propionate ratio after both additives’ supplementation. The expressions of PPARγ, SREBPF1, ACACA, FASN, SCD, and FABP3 mRNAs were enhanced by CCP or CT addition, but the relative expression of LPL mRNA was upregulated by CT addition only. Additionally, blood glucose, triglyceride, insulin-like growth factor-1, and total antioxidant capacity were promoted by both additives. The combination of CCP and CT more effectively increased the ruminal total VFA concentration, the acetate to propionate ratio, and blood glucose level, and decreased ammoniacal nitrogen concentration than that achieved by CCP or CT alone. The results suggested that CCP and CT supplementation stimulated lactation performance by promoting nutrient digestion and fatty acid synthesis in the mammary glands.

## 1. Introduction

It is well established that pantothenic acid (PA), being involved in the biosynthesis of coenzyme A and acyl-carrier protein, plays a crucial role in fatty acid, carbohydrate, and amino acid metabolisms [1]. Previous research found that supplementation of coated pantothenate (CCP) could promote the performance, nutrients digestion, and nitrogen balance in dairy bulls [2], and elevate total volatile fatty acid (VFA) content in the rumen, populations of fiber-degrading bacteria and the synthesis of microbial protein in beef cattle [3]. Other literature reported that the molar percentage of ruminal acetate was increased with PA supplements in the diets of dairy cows [4]. Furthermore, Wolin et al., found that pantothenate was needed for some amylolytic bacteria growth in the rumen [5].

Thiamine, as the component of thiamine pyrophosphate, is an essential cofactor in carbohydrate metabolism and is indispensable for normal cellular functions and growth [6]. Conventionally, the amount of thiamine synthesized by the ruminal microbiota is adequate to meet the ruminant’s needs. However, thiamine synthesized by rumen microorganisms could not meet the needs of host animals due to the improvement of breeds, intensive feeding, and continually improving production levels. Moreover, previous studies found that thiamine deficiency appeared when ruminants had ruminal acidosis [7]. The addition of thiamine in a high-concentrate (HC) diet was reported to positively influence dry matter (DM) intake and milk production [8]; it also increased ruminal pH [7], percentages of milk fat and protein [9], ruminal acetate concentration and the ratio of acetate to propionate, fungal abundance, and decreased ruminal ammoniacal nitrogen concentration [8] and rumen lactate concentration [7]. However, Xue et al., found that milk protein content was not altered by added thiamine [8]. Furthermore, Pan et al., found that thiamine addition decreased ruminal VFA concentration and increased ruminal pH in dairy cows that were administered an HC diet [7]. Nevertheless, how thiamine affects lactation performance under a normal diet has not yet been reported. Moreover, approximately 68% of thiamine provided as dietary supplements would be degraded in the rumen [10]. Therefore, it is necessary to study the impacts of coated thiamine (CT) on the lactation performance of dairy cows.

The above studies indicated that dietary PA or thiamine supplementation was helpful for ruminal microbial growth and nutrient digestion. Moreover, both PA and thiamine play dominant roles in intermediate energy metabolism. Thiamine plays a key role as a cofactor in glycolytic metabolism, especially associated with the five-enzyme complex pyruvate dehydrogenase (PDH), which catalyzes pyruvate as acetyl CoA into the tricarboxylic acid cycle [11]. The PA, being a crucial part of CoA biosynthesis, plays an important role in carbohydrate, amino acid, and fatty acid metabolisms [12]. Considering the above analysis, we hypothesized that the combination of PA and thiamine would be more effective in obtaining high milk yields through stimulated rumen fermentation, nutrient digestion, and fatty acid synthesis of the bovine mammary gland than PA or thiamine alone. Hence, this study aims to evaluate the influences of CCP and CT additives on milk production and milk composition, nutrient digestion, rumen fermentation, blood parameters, and expressions of genes involved in milk fatty acid synthesis by the bovine mammary gland.

## 2. Materials and Methods

### 2.1. Animal Management, Experimental Design, and Diets

This study was conducted from May 2021 to August 2021 at a commercial dairy farm (Shanxi Yinshanhu Dairy Cow Breeding Co. Ltd., Xinzhou, China). Forty-four multiparous Chinese Holstein cows (2.8 ± 0.19 of parity, 772 ± 12.3 kg of body weight [BW], 65.8 ± 8.6 days in milk [DIM] and 35.3 ± 1.9 kg/d of milk production, mean ± SD) were blocked by parity, BW, DIM, and milk production, and they were allocated four treatments in a 2 × 2 factorial block design, three replicates per group, and 3, 4 and 4 cows for one of three replicates, respectively). Additional CCP (0 mg/kg [CCP−] or 55 mg/kg dry matter [DM] of calcium pantothenate from CCP [CCP+]) and CT (0 g/kg [CT−] or 5.3 mg/kg DM of thiamine from CT [CT+]) were hand-mixed into the top one-third of total mixed ration (TMR). CCP (100 g/kg of calcium pantothenate) and CT (20 g/kg of thiamine) additives were prepared following a method reported by Wang et al. [13]. The supplement of CCP contained 100 g/kg of calcium pantothenate, 500 g/kg of hydrogenated fat (ratio of C16:0–C18:0 = 2:1), 230 g/kg of silicon dioxide and 170 g/kg of calcium stearate. The supplement of CT contained 20 g/kg of thiamine, 550 g/kg of hydrogenated fat (ratio of C16:0–C18:0 = 2:1), 250 g/kg of silicon dioxide, and 180 g/kg of calcium stearate. The ruminal degradation rates of CCP and CT were 0.21 and 0.22, respectively, which were measured using rumen-cannulated cows. The disappearance rates of CCP and CT in the small intestine were 0.67 and 0.69, respectively, in duodenum-cannulated cows. Basal diets of lactating dairy cows (Table 1) were formulated based on the nutrient requirements of dairy cattle [14]. The feeding experiment was conducted for 105 days, including 15 days of the adaptation period followed by 90 days of the sampling period. Cows were housed in a naturally ventilated, two-row head-to-head free-stall barn, milked three times per day (at 06:30, 14:30, and 21:30 h), fed the same standard diet ad libitum at 08:00 h and 17.30 h daily, and free access to water was provided.

### 2.2. Measurement and Sample Collection of Feed and Milk

During the sampling period, all cows were weighed at 16:00 h on day 1 and day 90. The TMR offered and refused were recorded for each dairy cow daily to calculate DM intake, and subsequently, samples of TMR and refusals were sampled once every ten days and kept at −20 °C for future investigation. Milk produced by each dairy cow was measured daily. Milk was sampled at each milking daily and stored at 4 °C with 2-bromo-2-nitropropane-1, 3-diol for later examination.

### 2.3. Ruminal Fermentation

The rumen fluid was collected from each cow through an oral stomach tube at 7:00, 10:00, 13:00, and 16:00 h on days 44 and 89 of the sampling period. To avoid saliva contamination, the initially collected ruminal fluid of about 150 mL was discarded, and the following 200 mL fluid was reserved. Ruminal pH for all cows was measured simultaneously by a portable pH meter (ST3100/F, Zhejiang Nader Scientific Equipment Co. Ltd., Hangzhou, China). Samples of rumen fluid were filtered by four layered pieces of medical gauze. For subsequent analysis of VFA, 5 mL filtrate was mixed with 1 mL of 250 g/L meta-phosphoric acid and stored at −20 °C. Similarly, 5 mL filtrate was mixed with 1 mL of 20 g/L (*w*/*v*) sulfuric acid and stored at −20 °C for ammoniacal nitrogen testing. 

### 2.4. Total Tract Apparent Nutrient Digestion Assay

At 06:30 and 18:30 h during experimental days 70–87, all cows were dosed with 5 g of chromic oxide powder, which was placed in a gelatin capsule and considered as a digestion marker. During days 78–87, about 250 g of fecal samples were collected from each cow’s rectum at 7:00, 13:00, 19:00, and 1:00 h and kept at −20 °C. During days 88–89, samples of the feed, refusals, and feces were composited by the cow, dried at 55 °C for 48 h, and ground to pass through a 1 mm screen with a cutter mill (Beishengwei Experimental Instrument Co., Ltd., Changzhou, China).

### 2.5. Blood Metabolites

At 10:30 h on day 90 of the sampling period, a sample of blood was collected from the coccygeal vessel of cows using 10 mL evacuated tubes (Hunan Liuyang Medical Instrument Co. Ltd., Liuyang, China). To separate serum, samples of blood were centrifuged at 2000× *g* and 4 °C for 12 min. Serum samples were stored at −20 °C.

### 2.6. Mammary Tissue Sampling

On day 90 of the sampling period, mammary tissues were biopsied from 16:00 to 17:30 h. About 1000 mg of secretory tissues of mammary glands from each dairy cow was collected via surgical biopsy from the midpoint section of a rear quarter, as described by Farr et al. [15]. Mammary tissue biopsy samples were quickly frozen using liquid nitrogen and then kept at −80 °C until total tissue RNA extraction.

### 2.7. Chemical Analyses

The samples of feed, refusal, and feces were analyzed for the DM content (method 934.01), nitrogen (method 976.05), crude ash (method 942.05), and ether extract (EE) (method 973.18) based on AOAC [16]. Organic matter (OM) content was estimated as the difference between DM and crude ash. The neutral detergent fiber (NDF) was measured according to the procedure reported by Van Soest et al., using heat-stable alpha-amylase and sodium sulfite [17]. The acid detergent fiber (ADF) was analyzed based on the method of AOAC (method 973.18) [16]. The fat content, true protein, and lactose were analyzed using a Milko Scan FT-120 unit (Foss Electric, Hillerød, Denmark) according to the method of AOAC (method 972.16) [16]. Fecal chromium content was estimated using atomic absorption spectrophotometry. 

Gas chromatographic measurement of fatty acid composition in milk was conducted (Agilent 7890 A, Agilent Technologies, Santa Clara, CA, USA) using an autosampler, flame ionization detector, and split injection. Milk was centrifuged to obtain the milk fat cake, milk fat was extracted, and trans-methylation of the esterified FA was performed according to a previous report [18]. 

Concentrations of ruminal VFA were measured by gas chromatography (GC7800; Beijing Purui Analytical Instrument Co. Ltd., Beijing, China). Ammoniacal nitrogen content was analyzed according to the procedure of AOAC [16]. Serum glucose, total protein, albumin, triglyceride, insulin-like growth factor-1 (IGF-1), pantothenate, thiamine, and total antioxidant capacity (T-AOC) were analyzed by the Konelab TM auto-analyzer (Thermo Fisher Scientific Oy, Vantaa, Finland) by using enzyme-linked immunosorbent assay test assay kit (Beijing Meilian Biology Science & Technology Co. Ltd., Beijing, China).

### 2.8. RNA Extraction from the Mammary Gland and Quantitative Real-Time PCR Analysis

The total RNA isolation kit (Invitrogen, Carlsbad, CA, USA) was used to extract total RNA from 100 mg mammary gland tissues according to the protocol provided by the manufacturer. The concentration and quality of the extracted RNA were analyzed using a NanoDrop ND-1000 Spectrophotometer (NanoDrop Technologies, Rockland, DE, USA). The ratios of absorbances recorded at 260 and 280 nm of all RNA samples were approximately 2.0. The integrity of the RNA was visually checked by denaturing agarose gel electrophoresis followed by ethidium bromide staining. According to the instructions provided by the manufacturer, the iScriptTM cDNA Synthesis Kit (Bio-Rad Laboratories GmbH, Munich, Germany) was used to synthesize cDNA from 500 ng of each total RNA sample in 10 μL reaction volume. The reaction conditions were as follows: 15 min at 37 °C and 5 min at 85 °C. Negative control reactions were carried out without reverse transcriptase for each sample to detect possible contamination by genomic DNA or environmental DNA.

The abundances of sterol regulatory element-binding factor 1 (SREBPF1), peroxisome proliferator-activated receptor γ (PPARγ), fatty acid synthase (FASN), acetyl-coenzyme A carboxylase-α (ACACA), fatty acid-binding protein 3 (FABP3), stearoyl-CoA desaturase (SCD), and lipoprotein lipase (LPL) mRNAs were quantified by qRT-PCR using the iCycler and the iQ-SYBR Green Supermix (Bio-Rad Laboratories, Hercules, CA, USA). The primer sets used for real-time PCR are shown in Table 2. Subsequent qPCR was performed on a MxPro-Mx3000P multiplex quantitative PCR system (Stratagene, La Jolla, CA, USA) in triplicate. A reaction mixture of 20 μL comprised 2 μL cDNA, 10 µL SYBR Premix TaqTM Ⅱ (TaKaRa Biotechnology Co., Ltd., Dalian, China), 0.8 µL PCR Forward Primer (10 μM), 0.8 µL PCR Reverse Primer (10 μM), 0.4 µL ROX Reference Dye Ⅱ (TaKaRa Biotechnology Co., Ltd., Dalian, China), and 6.0 µL dH_2_O. The following PCR cycling conditions were applied: 1 cycle of 95 °C for 20 s, 45 cycles of 95 °C for 20 s, annealing temperature for 30 s, and 62 °C for 20 s, followed by a melting curve analysis. Each denaturation and extension step was followed by fluorescence detections. The relative mRNA quantity of ACACA, FASN, SCD, PPARγ, SREBF1, FABP3, and LPL was quantified in reference to 18S rRNA according to the following equation: relative quantification = 2^−(Ct target genes − Ct total reference genes)^, where Ct represents threshold cycle.

### 2.9. Statistical Analyses

Fat-corrected milk (FCM) was estimated as 0.4 × milk yield + 15 × fat yield. Energy-corrected milk (ECM) was estimated as 0.3246 × milk yield + 12.86 × fat yield + 7.04 × protein yield [14]. The feed efficiency (FE) was estimated as the ratio of milk yields (milk and ECM yields) to the DM intake for each dairy cow. Data of DM intake, milk performance, and FE for statistical analysis were the mean of that recorded every 30 days. The SAS procedures (Proc Mixed; SAS, 2002) were used to analyze all data in a 2 (CCP additive) × 2 (CT additive) randomized block design; the statistic model was as follows:*Y_ijklm_* = *μ* + *B_i_* + *C_j_* + *D_k_* + (*CD*) *_jk_* + *T _l_* + *(TC) _jl_* + *(TD) _kl_* + *(TCD) _jkl_* + *R_m:ijk_* + *ε_ijklm_*

Data for digestibility coefficients, rumen fermentation parameters, composition of milk fat, and blood parameters were also analyzed by the above model. Data for the expression of genes related to fatty acid synthesis in the mammary gland was analyzed as follows:*Y_ijklm_* = *μ* + *B_i_* + *C_j_* + *D_k_* + (*CD*) *_jk_* + *R_m:ijk_* + *ε_ijklm_*
where, *Y_ijklm_* represents the dependent variable; *μ* represents the overall mean, *B_i_* represents the random effects of the ith block, *C_j_* represents the fixed effects of CCP additive (*j* = 0 or 55 mg/kg DM of pantothenate), *D_k_* represents the fixed effects of CT additive (*k* = 0 or 5.3 mg/kg DM of thiamine), (*CD)_jk_* represents the CCP×CT interaction; *T_l_* represents the fixed effect of time; *(TC)_jl_* represents the time × CCP interaction; *(TD)_kl_* represents the time × CT interaction; *(TCD)_jkl_* represents the time × CCP × CT interaction; *R_m_* represents the random effects of the mth cow; and ε*_ijklm_* represents the residual error. The covariance structure was first-order autoregressive. For rumen fermentation parameters, microbial enzymatic activity, and microflora, sampling time was considered as repeated measurements. Mean separations using the probability of difference tests (PDIFF in SAS) were considered only for effects that were significant (*p* < 0.05). The significant differences were declared at *p* < 0.05. 

## 3. Results

### 3.1. DM Intake, Milk Yields, Milk Composition and Feed Efficiency

The CCP × CT, time × CCP, time × CT, and time × CCP × CT interactions were not significant for DM intake, production, milk composition, and feed efficiency (Table 3). The DM intake was higher (*p* < 0.05) for CCP or CT supplementation. Productions of the actual milk, fat-corrected milk, energy-corrected milk, fat, true protein, and lactose contents were higher (*p* < 0.05) in treatment sets of both supplements compared with that of the control. Milk fat percentage was enhanced (*p* = 0.037) by CT additive but was unaltered in the sets with CCP additive. None of the supplements altered the percentages of milk true protein and lactose, whereas both enhanced the feed efficiency (*p* < 0.05).

Amounts of C11:0, C12:0, C13:0, C14:0, C15:0, C16:0, C24:0 fatty acids, and saturated fatty acids (SFA) in milk fat were increased (*p* < 0.05; Table 4) due to CCP or CT addition, but other SFAs were not influenced by CCP or/and CT additives. CCP or/and CT additives induced an increase in C14:1, C15:1, C16:1, C24:1 fatty acids contents, unsaturated fatty acid (UFA), and C4-16 fatty acids (*p* < 0.05; Table 5), whereas, they caused decreased C17-22 fatty acid contents (*p* < 0.05); however, other UFAs were not impacted by CCP or/and CT. The ratio of C14:1 to C14:0 (3.79%, 4.15%, 4.72%, and 4.99% for CCP−CT−, CCP−CT+, CCP+CT−, and CCP+CT+ treatments, respectively) was increased (*p* < 0.05) by CCP or/and CT additives. 

### 3.2. Digestibility Coefficient and Rumen Fermentation

Digestibility coefficients of DM, OM, crude protein (CP), NFC, NDF, and ADF were greater (*p* < 0.05; Table 6) as a result of treatments with both additives. However, the digestibility coefficient of EE was higher (*p* = 0.024) for the CCP additive but was not impacted by the CT additive. The significant effect of CCP × CT interaction was not detected on ruminal pH and percentages of acetate, propionate, butyrate, valerate, isobutyrate, and isovalerate; however, the total VFA content was significantly affected, the ratios of acetate to propionate and acetate molar percentage increased (*p* < 0.05) by the influence of CCP or CT additive. However, the increased magnitude was higher when only CT was used as a diet supplement compared with that achieved by only CCP additive. Ammoniacal nitrogen concentrations decreased (*p* < 0.05) by CCP or CT additive; however, the decreased magnitude was higher when CT and CCP were used as diet supplements compared to that in the case of only a CT-supplemented diet without CCP additive. Ruminal pH and propionate molar percentage were decreased (*p* < 0.05) by both additives. Dietary CCP or CT supplementations did not impact percentages of butyric, valeric, isobutyric, and isovaleric acids in dairy cows. 

### 3.3. Blood Metabolites

The time × CCP, time × CT, and time × CCP × CT interaction did not remarkably affect microbiota (*p* > 0.05; Table 7). The CCP × CT interaction remarkably (*p* = 0.012) affected serum glucose content which was increased (*p* < 0.05) with CCP or CT supplementation; however, it was more increased magnitude by CT supplement only compared to that by diets with both CT and CCP additives. Serum total protein and albumin concentration were increased (*p* < 0.05) by the inductive effect of CCP additive but were not impacted by CT additive. CCP or CT supplementation induced elevation of serum IGF-1 and T-AOC contents (*p* < 0.05). CCP supplementation increased serum pantothenate concentration (*p* = 0.012), and the thiamine content of blood was increased (*p* = 0.025) after applying the CT additive.

### 3.4. The Relative mRNA Expressions of Genes Involved in Milk Fatty Acids Synthesis

No significant effect of time × CCP, time × CT, and time × CCP × CT interaction was detected on genes expression (*p* > 0.05; Table 8). The expression of SREBPF1, PPARγ, ACACA, FASN, SCD, and FABP3 mRNAs was enhanced (*p* < 0.05) by CCP or CT addition. However, the relative expression of LPL mRNA was increased (*p* < 0.05) by the CT additive but was not impacted by the CCP additive.

## 4. Discussion

The increased DM intake induced by CCP or/and CT addition was attributed to the increased NDF digestibility coefficient since it was clarified that the reticulorumen distension could be mitigated by the increased dietary fiber digestibility [19]. The CCP− or/and CT-induced increased actual milk production, yields of FCM and ECM, fat, true protein, and lactose contents resulted from the comprehensive effect of the increased DM intake, dietary nutrients digestibility coefficient, and rumen total VFA production. Furthermore, the increased milk yields might also be associated with the promoted energy utilization after CCP or/and CT addition; pantothenate plays an important role in the utilization of energy derived from carbohydrates, fatty acids, and protein through coenzyme A, while thiamine plays an important role in carbohydrate metabolism for dairy cows [6]. Similarly, our previous experiments found positive influences of CCP on the DM intake of dairy cows, milk production, and composition of milk [13]. Previous reports detected a positive effect of thiamine on DM intake and milk yields in dairy cows based on an HC diet [8]. The present results showing increased yield and content of milk fat are consistent with that reported by Pan et al., who found that added thiamine in the HC diet increased synthesis and total amount of milk fat [9]. The unchanged milk protein content is parallel to the findings of Xue et al., who detected no impact of thiamine supplementation on milk protein content [8]. 

The elevation of C11:0, C12:0, C13:0, C14:0, C15:0, and C16:0 fatty acid contents with CCP+ or/and CT+ can be attributed to the increased ruminal acetate and butyrate concentrations. Principally, de novo synthesis of milk fatty acids C4-16 occurs from acetic and β-hydroxybutyric in the bovine mammary gland [20], and β-hydroxybutyric is synthesized from butyric via ketogenesis in ruminal epithelium cell [21]. Thus, the increased acetic acid and butyric acid with CCP or/and CT addition provided sufficient precursors for the de novo synthesis of milk fat and increased milk fat yield. Furthermore, the infusion of acetate in the rumen increased milk fat percentage and the C4:0 and C16:0 contents [22]. Additionally, in vitro studies also detected a positive influence of acetate and β-hydroxybutyrate on fat and protein synthesis [20]. Although the C17-22 fatty acid content in milk decreased after CCP or/and CT addition, its production remained unaltered. These results indicated that the absorption and transport of long-chain fatty acids (LCFA) was not increased by CCP or CT supplements. The C14:1 to C14:0 ratio in milk reflects the SCD activity in the bovine mammary glands [23]; hence, the increased C14:1 to C14:0 ratio with CCP+ or/and CT+ indicated the desaturation of SFA in the mammary gland of dairy cow caused by experimental supplementation of the diet.

The increased digestibility coefficient of DM, OM, CP, EE, NFC, NDF, and ADF with CCP+ was caused by the improvement in microbial enzymatic activity and altered ruminal microbiota, and implied that ruminal nutrient degradation was promoted by CCP supplement. It was confirmed that ruminal degradations of DM, CP, and NDF were enhanced by CCP additive in steers [3]. The greater digestibility coefficients of DM, OM, CP, NDF, ADF, and NFC induced by CT might be partly due to the elevation in microbial enzymatic activity and bacteria populations, and suggested that ruminal nutrient degradation was promoted by CT supplements. Although calcium pantothenate and thiamine were coated, some of them were degraded in the rumen, promoted microbial growth, and increased microbial enzymatic activity. Contrarily, the increased nutrient digestibility coefficient might be due to the increased activity of intestinal digestive enzymes promoted by CCP or/and CT additives; however, further research is needed to confirm this correlation. 

The reduction of ruminal pH with CCP+ or/and CT+ is in keeping with the greater total VFA content, which resulted from the increased DM intake, microbial populations, and enzymatic activities after diet supplementation. The increased acetate-to-propionate ratio promoted by CCP or/and CT additives implied that the ruminal fermentation pattern tended to form more acetate. Following CCP application, acetate concentration was enhanced by 6.53% (80.25 and 85.45 mM for CCP− and CCP+), and propionate concentration increased by 3.46% (29.35 and 30.40 mM for CCP− and CCP+). CT additives enhanced acetate and propionate concentrations by 10.68% (78.65 and 87.05 mM for CT− and CT+) and 3.23% (29.41 and 30.36 mM for CT− and CT+), respectively. These results were ascribed to the positive influences of supplementary CCP or/and CT on microbial growth and were parallel with the increased digestibility of NFC and NDF. Similarly, literature demonstrated that ruminal VFA concentration, and the acetate-to-propionate ratio were promoted by CCP supplements in steers [3] and in dairy cows [13]. Other reports indicated that the addition of pantothenate increased acetate molar percentage in dairy cows [4]. The literature demonstrated that thiamine supplements in dairy cows fed an HC diet increased ruminal acetate content, and the ratio of acetate to propionate [8]. However, Pan et al., demonstrated that thiamine supplement decreased ruminal VFAs concentration, and increased ruminal pH in dairy cows fed with an HC diet [7]. Ammoniacal nitrogen is mainly derived from degraded dietary proteins and used for the synthesis of microbial proteins; the decreased ammoniacal nitrogen concentration implies that bacterial protein synthesis was stimulated [8]. The results can be confirmed by the urinary excretion of total purine derivatives, which was enhanced by CCP additive [2].

The CCP × CT interaction significantly affected ruminal total VFA content, the acetate-to-propionate ratio, and ammoniacal nitrogen content; the increased magnitude of total VFA content, and the acetate-to-propionate ratio were detected to be higher after adding CCP in diets supplemented with CT additives than that achieved by diets without CT additive. The combination of CCP and CT decreased ammoniacal nitrogen content more than that induced by CCP or CT alone. These findings suggested that calcium pantothenate and thiamine may have synergistic growth-promoting effects on ruminal microorganisms. Previous reports have demonstrated that ruminal cellulolytic bacteria and enzymatic activity are increased by CCP supplements in steers [3]. Thiamine is essential for microbial growth, such as *Ruminococcus albus* and *R. flavefasciens* [24,25]; it serves as a cofactor of phosphoenolpyruvate decarboxylase in *Bacteroides fragilis* [26] and takes part in the formation of acetyl-CoA via pyruvate- ferredoxin oxidoreductase in *Megasphaera*, *Ruminococcus*, *Butyrivibrio fibrisolvens*, and *Selenomonas ruminantium* [27]. Similarly, thiamine supplements in the HC diet stimulated the proliferation of ruminal fungi in dairy cows [8]. Moreover, the mutualism of fungi and bacteria might be promoted by CT, and then result in increased CAZymes which degrade carbohydrates to monosaccharides and further hydrolyze into pyruvate [8]. 

The increased serum PA content suggested that PA absorption was promoted by CCP supplement, and it was evidenced by a previous study that reported that plasma PA content increased with 50, 100, or 200 mg/d of rumen-protected PA addition [28]. The increased blood thiamine content suggested that the absorption of thiamine was promoted by the CT additive. Blood glucose is mainly synthesized via the propionate gluconeogenesis pathway, and the supply of propionate from rumen fermentation was positively related to hepatic glucose synthesis [29]. The CCP− induced increased blood glucose was confirmed by the increased plasma glucose content in dairy cows [30]. The greater blood glucose content, which resulted from greater rumen propionate concentration, suggested that dietary CT supplements might improve gluconeogenic capacity in dairy cows. Total protein content in blood reflect protein availability [31]; the CCP− or/and CT -induced elevation of total protein content supported the increased milk yields, and implied that the availability of dietary protein was altered by CCP or/and CT additives. Similarly, blood albumin, globulin, and glucose concentrations were enhanced by 40 mg/d of thiamine in sheep [32]. However, thiamine supplemented at concentrations of 4 and 6 mg/kg of DM showed no influence on blood glucose content in lambs fed an HC diet [33]. The difference in thiamine on ruminant metabolism may be due to the different doses of thiamine additives and dietary compositions.

The CCP × CT interaction significantly affected blood glucose concentration, and the combination of CCP and CT showed a more intense elevation of blood glucose concentration than that induced by CCP or CT alone. Nevertheless, no significant effect of CCP × CT interaction was observed on nutrient digestion. The above findings suggested that the energy metabolism efficiency was promoted by CCP or/and CT additive. Thiamine, as a cofactor, has a critical role in numerous reactions of glycolytic metabolism. It especially participates in the five-enzyme complex PDH, which catalyzes pyruvate as acetyl CoA into the tricarboxylic acid cycle [11]. Pantothenic acid participates in the metabolism of carbohydrates, amino acids, and fatty acids [12], as it is required for the biosynthesis of CoA. Thus, the current research indicated that supplementation of thiamine in the form of CT probably stimulated the utilization efficiency of pantothenate from CCP during the energy metabolism, and led to improved blood glucose content when a combination of CCP and CT was used for diet supplementation compared to that recorded with CCP or CT supplementation alone. 

The expression of *FASN*, *ACACA*, *FABP3, LPL*, and *SCD* was adjusted by *SREBPF1* and *PPARγ* in the mammary gland of dairy cows [34]; the FASN and ACACA take part in de novo synthesis of fatty acids in milk. A positive correlation between the expressions of *ACACA* and *FASN* mRNAs and the synthesis of carbon 4-16 fatty acids was described by Bernard et al. [23]. Furthermore, in vitro studies detected the supplementation of acetate and butyrate to elevate the mRNA expression for *PPARγ*, *SREBPF1*, *FASN*, *ACACA*, *SCD*, *FABP3*, and *LPL* in the epithelial cells of the mammary gland [35]. Thus, the increased mRNA expression of *SREBPF1*, *PPARγ*, *FASN*, and *ACACA* might be partly due to the increased ruminal acetate and butyrate production, and CCP or CT supplementation can partly confirm the promoted milk fat yields and C4-15 fatty acids concentration. The SCD catalyzes the synthesis of monounsaturated fatty acids from SFA [20], and the C14:1 to C14:00 ratio reflects the SCD activity in the bovine mammary gland. After CCP or CT supplementation, the upregulated SCD expression contributed to the increased ratios of C14:1 to C14:00. Moreover, the increased concentrations of C14:1, C15:1, and C16:1 in milk further implied that the SCD activity was stimulated by CCP or CT additive. The LPL and FABP3 were associated with the absorption and transport of LCFA in the epithelial cells of the mammary gland [20]; the upregulated FABP3 and LPL mRNAs indicated that CCP or CT supplements promoted the absorption and transport of LCFA.

## 5. Conclusions

The CCP or CT supplements in diets of dairy cows potentially improve milk yield, milk fat content, and feed efficiency through stimulation of total-tract nutrient digestion, rumen fermentation, and milk fatty acids synthesis. These two additives collectively promoted ruminal fermentation and blood glucose concentration more effectively than that achieved by CCP or CT alone. Future research should focus on the molecular mechanism of interaction between these two additives.

## Figures and Tables

**Table 1 animals-13-02526-t001:** Ingredients and chemical composition of the basal diet (DM basis).

Ingredients	Contents [%]
Corn fodder	25.5
Alfalfa hay	11.8
Oat hay	12.7
Corn grain, ground	25.7
Wheat bran	6.0
Soybean meal	9.0
Rapeseed meal	2.5
Cottonseed cake	5.0
Calcium carbonate	0.5
Salt	0.5
Dicalcium phosphate	0.3
Mineral and vitamin premix ^1^	0.5
Chemical composition	
Organic matter	94.5
Crude protein	16.4
Ether extract	3.25
Neutral detergent fiber	31.4
Acid detergent fiber	19.5
Non-fiber carbohydrate ^2^	43.5
Calcium	0.71
Phosphorus	0.45

^1^ Contained per kg premix: 20,000 mg Fe, 1600 mg Cu, 8000 mg Mn, 71,220 mg Zn, 120 mg I, 60 mg Se, 20 mg Co, 820,000 IU vitamin A, 300,000 IU vitamin D, and 10,000 IU vitamin E. ^2^ Non-fiber carbohydrate (NFC), calculated by 1000-CP-NDF-Fat-Ash.

**Table 2 animals-13-02526-t002:** The primers used in real-time PCR.

Gene	Primer Sequence (5’-3’)	GenBank Accession No.	Annealing Temperature (°C)	Size (bp)
*ACACA*	F: CATCTTGTCCGAAACGTCGAT R: CCCTTCGAACATACACCTCCA	AJ132890	58	101
*FASN*	F: AGGACCTCGTGAAGGCTGTGA R: CCAAGGTCTGAAAGCGAGCTG	NM001012669	62	85
*SCD*	F: TCCTGTTGTTGTGCTTCATCC R: GGCATAACGGAATAAGGTGGC	AY241933	58	101
*PPARγ*	F: AACTCCCTCATGGCCATTGAATG R: AGGTCAGCAGACTCTGGGTTC	NM181024.2	60	323
*SREBF1*	F: CTGACGACCGTGAAAACAGA R: AGACGGCAGATTTATTCAACTT	NM001113302	60	334
*FABP3*	F: GAACTCGACTCCCAGCTTGAA R: AAGCCTACCACAATCATCGAAG	DN518905	60	102
*LPL*	F: ACACAGCTGAGGACACTTGCC R: GCCATGGATCACCACAAAGG	BC118091	60	101
*GAPDH*	F: CCTGGAGAAACCTGCCAAGT R: AGCCGTATTCATTGTCATACCA	NM001034034.2	59	215

*ACACA* = acetyl-coenzyme A carboxylase- α; *FASN* = fatty acid synthase; *FABP3* = fatty acid-binding protein 3; *GAPDH* = glyceraldehyde-3-phosphate dehydrogenase; *LPL*= lipoprotein lipase; *PPARγ* = peroxisome proliferator-activated receptor γ; *SCD* = stearoyl-CoA desaturase; *SREBF-1* = sterol regulatory element-binding factor 1.

**Table 3 animals-13-02526-t003:** Effects of coated calcium pantothenate (CCP) and coated thiamine (CT) supplements on dry matter intake, lactation performance, and feed efficiency of lactating dairy cows.

Item	CCP− ^1^		CCP+			*p*-Values ^2^		
	CT−	CT+	CT−	CT+	SEM	CCP	CT	CCP × CT
DM intake (kg/d)	22.3	22.8	23.2	24.0	0.17	0.012	0.008	0.060
Milk production (kg/d)								
Actual	35.6	39.2	38.5	42.5	0.36	0.002	0.009	0.17
Fat-corrected milk ^3^	32.1	35.9	35.2	39.9	0.39	0.004	0.005	0.13
Energy-corrected milk ^4^	34.6	38.6	37.7	41.0	0.38	0.006	0.008	0.12
Fat	1.19	1.35	1.32	1.40	0.019	0.006	0.011	0.16
True protein	1.10	1.21	1.16	1.30	0.013	0.004	0.009	0.62
Lactose	1.83	2.03	1.93	2.20	0.020	0.003	0.006	0.37
Milk composition (g/kg)								
Fat	3.33	3.45	3.42	3.59	0.034	0.089	0.037	0.71
True protein	3.08	3.09	3.13	3.06	0.019	0.85	0.41	0.30
Lactose	5.15	5.16	5.23	5.19	0.017	0.15	0.61	0.42
Feed efficiency (kg/kg)								
Milk/DM intake	1.59	1.71	1.66	1.77	0.003	0.006	0.007	0.23
ECM/DM intake	1.55	1.69	1.62	1.71	0.002	0.008	0.011	0.19

^1^ CCP−, CCP+, CT− and CT+ with 0 mg CCP, 55 mg CCP, 0 mg CT, and 5.3 mg CT per kg dietary DM. ^2^ CCP: CCP effect; CT: CT effect; CCP × CT: the interaction between CCP and CT addition. The *p*-value of time for all variables was significant (*p* < 0.050). The time × CCP, time × CT, and time × CCP × CT interaction for all the studied variables were not significant (*p* > 0.050). ^3^ Fat-corrected milk = 0.4 × milk yield + 15 × fat yield. ^4^ Energy corrected milk (ECM) = 0.3246 × milk yield + 12.86 × fat yield + 7.04 × protein yield.

**Table 4 animals-13-02526-t004:** Effects of coated calcium pantothenate (CCP) and coated thiamine (CT) supplements on saturated fatty acid (FA) composition of milk fat in lactating dairy cows (g/kg fat).

Item	CCP− ^1^		CCP+			*p*-Values ^2^		
	CT−	CT+	CT−	CT+	SEM	CCP	CT	CCP × CT
C4:0	20.1	21.2	21.4	22.3	0.25	0.41	0.39	0.62
C6:0	21.5	22.6	22.7	23.6	0.36	0.70	0.51	0.33
C8:0	12.3	13.3	13.2	14.4	0.15	0.31	0.41	0.62
C10:0	25.9	26.7	26.1	27.5	0.47	0.38	0.38	0.53
C11:0	2.9	3.6	3.8	4.5	0.05	0.019	0.027	0.30
C12:0	33.3	35.3	36.1	38.1	1.05	0.047	0.031	0.61
C13:0	2.8	3.5	3.6	3.8	0.09	0.029	0.031	0.15
C14:0	110.1	116.6	118.7	126.2	1.06	0.041	0.030	0.051
C15:0	24.7	26.0	26.5	28.4	1.75	0.039	0.043	0.18
C16:0	57.8	62.1	61.5	63.2	1.85	0.045	0.038	0.55
C17:0	20.2	20.0	20.1	20.8	0.38	0.42	0.52	0.57
C18:0	71.2	70.3	73.2	70.1	2.24	0.19	0.24	0.86
C20:0	5.3	5.2	5.1	4.9	0.10	0.30	0.45	0.69
C21:0	0.8	0.75	0.8	0.9	0.02	0.47	0.53	0.99
C22:0	1.4	1.3	1.4	1.4	0.03	0.52	0.62	0.75
C23:0	0.6	0.6	0.6	0.7	0.02	0.70	0.65	0.84
C24:0	1.2	1.4	1.5	1.5	0.03	0.045	0.036	0.46
SFA ^3^	412.15	430.55	436.29	452.35	9.32	0.026	0.037	0.41

^1^ CCP−, CCP+, CT− and CT+ with 0 mg CCP, 55 mg CCP, 0 mg CT, and 5.3 mg CT per kg dietary DM. ^2^ CCP: CCP effect; CT: CT effect; CCP × CT: the interaction between CCP and CT addition. The *p*-value of time, the time × CCP, time × CT, and time × CCP × CT interaction for all the studied variables were not significant (*p* > 0.050). ^3^ SFA = saturated fatty acid.

**Table 5 animals-13-02526-t005:** Effects of coated calcium pantothenate (CCP) and coated thiamine (CT) supplements on unsaturated fatty acid (FA) composition of milk fat in lactating dairy cows (g/kg fat).

Item	CCP− ^1^		CCP+			*p*-Values ^2^		
	CT−	CT+	CT−	CT+	SEM	CCP	CT	CCP × CT
C14:1	4.17	4.8	5.6	6.3	0.84	0.002	0.004	0.068
C15:1	5.56	6.0	6.21	6.7	0.01	0.041	0.033	0.39
C16:1	74.5	77.8	78.0	83.4	1.55	0.010	0.039	0.078
C18:1	225.0	213.0	204.0	208.0	7.28	0.081	0.079	0.89
C20:1	2.4	2.7	2.6	3.1	0.05	0.58	0.31	0.10
C22:1	0.7	0.8	0.8	0.9	0.02	0.103	0.19	0.20
C24:1	0.7	0.9	0.9	1.1	0.02	0.032	0.021	0.085
C18:2 (cis-9,12)	116.2	109.0	110.0	95.6	2.23	0.059	0.061	0.317
C18:2 (trans-9,12)	104.0	103.8	102.7	91.4	1.62	0.10	0.12	0.20
C18:2 (cis-9,11)	5.5	5.7	5.3	5.3	0.23	0.52	0.60	0.71
C18:2(trans-10, cis-12)	9.4	9.3	9.2	9.4	0.37	0.49	0.51	0.82
C18:3 (cis-6,9,12)	1.6	1.4	1.7	1.5	0.04	0.70	0.82	0.99
C18:3 (cis-9,12,15)	11.3	11.2	11.1	10.2	0.23	0.50	0.60	0.14
C20:2	1.4	1.5	1.5	1.5	0.03	0.64	0.58	0.23
C20:3 (cis-8,11,14)	6.6	5.9	6.1	5.4	0.13	0.43	0.54	0.081
C20:3 (cis-11,14,17)	0.4	0.4	0.5	0.5	0.01	0.39	0.47	0.59
C20:4	9.1	9.6	10.1	9.2	0.18	0.54	0.62	0.19
C20:5	1.1	1.1	1.3	1.3	0.03	0.71	0.97	0.89
C22:2	0.3	0.3	0.3	0.5	0.01	0.11	0.061	0.11
C22:3	0.3	0.4	0.3	0.5	0.01	0.23	0.31	0.13
C22:4	2.2	2.5	2.6	2.6	0.05	0.10	0.098	0.99
C22:5 (cis-4,7,10,13,16)	0.2	0.2	0.3	0.2	0.01	0.47	0.50	0.10
C22:5 (cis-7,10,13,16,19)	2.7	2.8	2.9	2.8	0.06	0.53	0.48	0.99
C22:6	0.5	0.7	0.6	0.5	0.02	0.19	0.35	0.18
UFA ^3^	587.8	569.7	563.6	547.9	14.28	0.024	0.036	0.57
C4-16	397.6	417.6	423.3	448.4	10.33	0.029	0.038	0.63
C17-24	602.3	582.65	576.52	551.8	11.46	0.031	0.031	0.52

^1^ CCP−, CCP+, CT− and CT+ with 0 mg CCP, 55 mg CCP, 0 mg CT, and 5.3 mg CT per kg dietary DM. ^2^ CCP: CCP effect; CT: CT effect; CCP × CT: the interaction between CCP and CT addition. The *p*-value of time, the time × CCP, time × CT, and time × CCP × CT interaction for all the studied variables were not significant (*p* > 0.050). ^3^ UFA = unsaturated fatty acid.

**Table 6 animals-13-02526-t006:** Effects of coated calcium pantothenate (CCP) and coated thiamine (CT) supplements on nutrient digestibility and ruminal fermentation in lactating dairy cows.

Item	CCP− ^1^		CCP+			*p*-Values ^2^		
	CT−	CT+	CT−	CT+	SEM	CCP	CT	CCP × CT
Digestibility coefficient								
Dry matter	0.665	0.701	0.703	0.731	0.002	0.013	0.023	0.23
Organic matter	0.684	0.715	0.717	0.744	0.002	0.022	0.018	0.57
Crude protein	0.742	0.778	0.769	0.791	0.004	0.034	0.003	0.45
Ether extract	0.757	0.788	0.815	0.832	0.011	0.024	0.25	0.74
Neutral detergent fiber	0.561	0.598	0.605	0.640	0.003	0.016	0.026	0.87
Acid detergent fiber	0.498	0.538	0.551	0.584	0.004	0.033	0.014	0.59
Non-fiber carbohydrate	0.797	0.817	0.813	0.835	0.004	0.042	0.012	0.93
Ruminal fermentation								
pH	6.90	6.43	6.67	6.36	0.025	0.006	0.015	0.14
Total VFA (mM)	118	132	130	137	0.379	0.024	0.032	0.027
Mol/100 mol								
Acetate	62.9	64.4	63.9	64.3	0.09	0.083	0.004	0.069
Propionate	24.1	22.6	23.4	22.0	0.08	0.008	0.006	0.36
Butyrate	10.3	10.1	9.8	10.6	0.05	0.94	0.10	0.12
Valerate	1.29	1.36	1.38	1.44	0.015	0.18	0.32	0.96
Isobutyrate	0.52	0.54	0.56	0.62	0.015	0.47	0.22	0.42
Isovalerate	0.77	0.82	0.83	0.87	0.021	0.21	0.30	0.93
Acetate:Propionate	2.62	2.86	2.75	2.92	0.012	0.005	0.007	0.046
Ammoniacal N (mg 100/mL)	13.2	11.6	12.4	9.5	0.13	0.008	0.009	0.010

^1^ CCP−, CCP+, CT− and CT+ with 0 mg CCP, 55 mg CCP, 0 mg CT, and 5.3 mg CT per kg dietary DM. ^2^ CCP: CCP effect; CT: CT effect; CCP × CT: the interaction between CCP and CT addition. The *p*-value of time was significant (*p* < 0.050) for the digestibility coefficient of crude protein, neutral detergent fiber, acid detergent fiber and non-fiber carbohydrate, total VFA, percentages of acetate and propionate, acetate to propionate ratio. The time × CCP, time × CT, and time × CCP × CT interaction for all the studied variables were not significant (*p* > 0.050).

**Table 7 animals-13-02526-t007:** Effects of coated calcium pantothenate (CCP) and coated thiamine (CT) supplements on blood metabolites of lactating dairy cows.

Item	CCP− ^1^		CCP+			*p*-Values ^2^		
	CT−	CT+	CT−	CT+	SEM	CCP	CT	CCP × CT
Glucose (mmol/L)	24.9	26.8	26.4	28.5	2.51	0.005	0.002	0.012
Total protein (g/L)	78.4	83.5	83.1	84.4	7.58	0.001	0.73	0.092
Albumin (g/L)	29.5	33.9	32.5	34.8	2.84	0.004	0.11	0.13
Triglyceride (mmol/L)	5.46	5.51	5.49	5.54	0.077	0.004	0.007	0.12
IGF-1 ^3^ (ng/mL)	282.4	313.1	308.3	316.3	2.89	0.012	0.019	0.28
T-AOC ^4^ (U/mL)	5.79	6.31	6.52	6.88	0.075	0.001	0.007	0.61
Pantothenate (μg/mL)	0.23	0.24	0.26	0.27	0.003	0.012	0.83	0.13
Thiamine (μg/L)	24.6	33.3	27.0	34.6	2.65	0.74	0.025	0.21

^1^ CCP−, CCP+, CT− and CT+ with 0 mg CCP, 55 mg CCP, 0 mg CT, and 5.3 mg CT per kg dietary DM. ^2^ CCP: CCP effect; CT: CT effect; CCP×CT: the interaction between CCP and CT addition. The *p*-value of time was significant (***p*** < 0.050) for IGF-1, T-AOC, pantothenate, and thiamine. The time × CCP, time × CT, and time × CCP × CT interaction for all the studied variables were not significant (*p* > 0.050). ^3^ IGF-1 = insulin-like growth factor-1. ^4^ T-AOC = total antioxidation capacity.

**Table 8 animals-13-02526-t008:** Effects of coated calcium pantothenate (CCP) and coated thiamine (CT) supplements on the mRNA expression of genes related to fatty acid synthesis in mammary glands of lactating dairy cows (proportion of 18S rRNA).

Item ^3^	CCP− ^1^		CCP+			*p*-Values ^2^		
	CT−	CT+	CT−	CT+	SEM	CCP	CT	CCP × CT
*PPARγ*	1.00	1.42	1.56	2.03	0.036	0.021	0.036	0.34
*SREBF1*	1.01	1.69	1.70	2.31	0.026	0.019	0.035	0.22
*ACACA*	1.00	1.45	1.53	2.05	0.029	0.036	0.048	0.22
*FASN*	1.00	1.59	1.56	2.26	0.027	0.018	0.015	0.54
*SCD*	1.05	1.50	1.55	2.07	0.017	0.023	0.034	0.26
*FABP3*	1.00	1.69	1.66	2.12	0.035	0.019	0.013	0.31
*LPL*	1.03	1.35	1.41	1.76	0.018	0.051	0.027	0.49

^1^ CCP−, CCP+, CT− and CT+ with 0 mg CCP, 55 mg CCP, 0 mg CT, and 5.3 mg CT per kg dietary DM. ^2^ CCP: CCP effect; CT: CT effect; CCP × CT: the interaction between CCP and CT addition. ^3^ *PPARγ* = peroxisome proliferator-activated receptor γ; *SREBF1* = sterol regulatory element- binding factor 1; *ACACA* = acetyl-coenzyme A carboxylaseα; *FASN* = fatty acid synthase; *SCD* = stearoyl-CoA desaturase; *FABP3* = fatty acid-binding protein 3; *LPL* = lipoprotein lipase.

## Data Availability

None of the data are deposited in an official repository. All data are available on request to the corresponding author.

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
