# Peer review of "Milk Yields and Milk Fat Composition Promoted by Pantothenate and Thiamine via Stimulating Nutrient Digestion and Fatty Acid Synthesis in Dairy Cows"

_animals, 2023, doi:10.3390/ani13152526_

Round 1

Reviewer 1 Report

In the manuscript ID animals-2474966, the authors examined the effects of pantothenate or/and thiamine supplement on milk production, nutrient digestibility, ruminal fermentation, blood metabolites, mRNA expression in mammary gland. This paper contains an original experiment demonstrating the combined effects of pantothenate and thiamine on the nutritional physiology of lactating dairy cows. The methodology for this experiment is fine. The conclusions are generally consistent with the results and discussion and correspond to the subject of the paper. However, the references and minor points could be improved. Please check the following:

Why is it called coated pantothenate and thiamine instead of rumen-protected pantothenate and thiamine? What is it coated with and what is the effect?

L56 Rewrite “Xue et al. 8”

L91 I think that the ruminal degradation rate differs from or is not stated in [13] Wang et al. Are the reference correct?

L211 Rewrite “μrepresents”

L227-229 What about energy-corrected milk?

L241 Rewrite “C16:00 C24:00”

L246-248 What is the increase rate of "The ratio of C14:1 to C14:0"?

L247 Rewrite “C14:00”

L269 Rewrite ”the ratios acetate : propionate”

L270 Rewrite “nitrogen content : total VFA content”

L271 “(p < 0.05) by the influence of CCP or CT additive” is not a same as Table 6.

L329 Rewrite “A, 1”

L330 Rewrite “cows.6”

L333 Rewrite “diet, 8”

L339-388 I think “CCP-induced” is “CCP+”, but readers might confuse it with “CCP-“.

L340 Rewrite “C16:00”

L359 Rewrite “steers, 3”

Author Response

We would like to thank you for providing helpful comments on our manuscript. We have attempted to address all concerns, as itemized below:

In the manuscript ID animals-2474966, the authors examined the effects of pantothenate or/and thiamine supplement on milk production, nutrient digestibility, ruminal fermentation, blood metabolites, mRNA expression in mammary gland. This paper contains an original experiment demonstrating the combined effects of pantothenate and thiamine on the nutritional physiology of lactating dairy cows. The methodology for this experiment is fine. The conclusions are generally consistent with the results and discussion and correspond to the subject of the paper. However, the references and minor points could be improved. Please check the following:

Why is it called coated pantothenate and thiamine instead of rumen-protected pantothenate and thiamine? What is it coated with and what is the effect?

AU: In previous publications, reviewers suggested that coated vitamins were more accurate than rumen-protected vitamins, because partly vitamins acted after release in the rumen. The supplement of CCP contained 100 g/kg of calcium pantothenate, 500 g/kg of hydrogenated fat (ratio of C16:0-C18:0=2:1), 230 g/kg of silicon dioxide and 170 g/kg of calcium stearate. The supplement of CT contained 20 g/kg of thiamine, 550 g/kg of hydrogenated fat (ratio of C16:0-C18:0=2:1), 250 g/kg of silicon dioxide and 180 g/kg of calcium stearate. The ruminal degradation rates of CCP and CT were 0.21 and 0.22, respectively, the disappearance rates of CCP and CT in the small intestine were 0.67 and 0.69, respectively.

L56 Rewrite “Xue et al. 8”

AU: Accepted, it has been revised as “Xue et al. found that milk protein content was not altered by added thiamine [8]. ”.

L91 I think that the ruminal degradation rate differs from or is not stated in [13] Wang et al. Are the reference correct?

AU: Accepted, The references are out of place, it has been revised as “additives were prepared following a method reported by Wang et al [13]. The ruminal degradation rates of CCP and CT were 0.21 and 0.22, respectively, which were measured using rumen cannulated cows. ”.

L211 Rewrite “μrepresents”

AU: Accepted, it has been revised as “μ represents”.

L227-229 What about energy-corrected milk?

AU: Accepted, it has been revised as “Productions of the actual milk, fat-corrected milk, energy-corrected milk, fat, true protein, and lactose contents were higher (p < 0.05) in treatment sets of both supplements compared with that of control. ”.

L241 Rewrite “C16:00 C24:00”

AU: Accepted, it has been revised as “C16:0, C24:0”.

L246-248 What is the increase rate of "The ratio of C14:1 to C14:0"?

AU: Accepted, it has been revised as “The ratio of C14:1 to C14:00 (3.79%, 4.15%, 4.72%, and 4.99% for CCP-CT-, CCP-CT+, CCP+CT-, and CCP+CT+ treatments, respectively) was increased (p < 0.05) by CCP or/and CT additives. ”.

L247 Rewrite “C14:00”

AU: Accepted, it has been revised as “C14:0”.

L269 Rewrite ”the ratios acetate : propionate”

AU: Accepted, it has been revised as “the ratios of acetate to propionate”.

L270 Rewrite “nitrogen content : total VFA content”

AU: Accepted, it has been revised as “nitrogen content to total VFA content”.

L271 “(p < 0.05) by the influence of CCP or CT additive” is not a same as Table 6.

AU: Accepted, the sentence “ruminal ammoniacal nitrogen content to total VFA content, as well as” has been deleted.

L329 Rewrite “A, 1”

AU: Accepted, it has been revised as “A, ”.

L330 Rewrite “cows.6”

AU: Accepted, it has been revised as “dairy cows [6]”.

L333 Rewrite “diet, 8”

AU: Accepted, it has been revised as “diet [8]”.

L339-388 I think “CCP-induced” is “CCP+”, but readers might confuse it with “CCP-“.

AU: Accepted, it has been revised as “The increased digestibility coefficient of DM, OM, CP, EE, NFC, NDF, and ADF with CCP+ ”.

L340 Rewrite “C16:00”

AU: Accepted, it has been revised as “C16:0”.

L359 Rewrite “steers, 3”

AU: Accepted, it has been revised as “steers [3]”.

Reviewer 2 Report

The manuscript is a complete investigation of the effect of supplementing pantothenate and thiamine on pre and post absorptive metabolism in the cows. The dietary treatments caused an increase in milk fat that appears to be related to modification of rumen digestion and fatty acid synthesis. The experiments were well done and the paper is sound as far as I can tell. Below minor revision can be made before publication:

Line 23 and other places: increased rather than elevated.

Line 23 and other places: decreased rather than reduced.

Line 89-91: It is unclear how was added. Was each cow fed and the treatment topdressed and hand mixed?

Line 107: For future work I would suggest multiple days of body weight since it can be variable from day to day.

Line 337: deleted.

Table 1: I prefer to report diets as a percent. The Journal may require following the convention of g/kg so you may need to keep it that way.

Table 4, Table 5 and Table 8: Example of significant digits. Only need two decimal place when P >0.1.

Author Response

We would like to thank you for providing helpful comments on our manuscript. We have attempted to address all concerns, as itemized below:

The manuscript is a complete investigation of the effect of supplementing pantothenate and thiamine on pre and post absorptive metabolism in the cows. The dietary treatments caused an increase in milk fat that appears to be related to modification of rumen digestion and fatty acid synthesis. The experiments were well done and the paper is sound as far as I can tell. Below minor revision can be made before publication:

Line 23 and other places: increased rather than elevated.

AU: Accepted, it has been revised according to your suggestion.

Line 23 and other places: decreased rather than reduced.

AU: Accepted, it has been revised as according to your suggestion.

Line 89-91: It is unclear how was added. Was each cow fed and the treatment topdressed and hand mixed?

AU: Accepted, it has been revised as “....were hand-mixed into the top one-third of total mixed ration (TMR)”.

Line 107: For future work I would suggest multiple days of body weight since it can be variable from day to day.

AU: Accepted, we would like to thank you for providing helpful comments.

Line 337: deleted.

AU: Accepted, it has been deleted.

Table 1: I prefer to report diets as a percent. The Journal may require following the convention of g/kg so you may need to keep it that way.

AU: Accepted, it has been revised as a percent according to your suggestion.

Table 4, Table 5 and Table 8: Example of significant digits. Only need two decimal place when P >0.1.

AU: Accepted, it has been revised according to your suggestion.

Reviewer 3 Report

The manuscript is well written. There are valuable results and information evaluating changing in milk yield and milk fat composition by adding CCP and CT to the feed.

Author Response

The manuscript is well written. There are valuable results and information evaluating changing in milk yield and milk fat composition by adding CCP and CT to the feed.

AU: Thank you very much for your evaluation of our manuscript, we have improved the result part.

Reviewer 4 Report

Suggestions:

·       Rows 84-85: how were animals grouped?

·       Row 97: when was diet administered?

·       Row 104: why were animals weighed at 16 h?

·       Rows 111,112,121,129,134,135: how were time sampling and days intervals established?

Minor revision:

·       Row 18: elevated milk

·       Row 81: from May

·       Row 94: What does NRC stand for?

·       Table 2: description on the same page of the table, please check the rest of the article

·       Row 211: μ represents

·       Table 3 in one page, please check the rest of the article, description is not clear, put a superscript per abbreviation

·       Row 244: UFA stands for?

·       Table 4: saturated fatty acid (SFA), description is not clear, put a superscript per abbreviation

·       Table 5: unsaturated fatty acid (UFA), description is not clear, put a superscript per abbreviation, please check the rest of the article

·       Row 264: what does CP stand for? Every abbreviation has to be explained

·       Row 320: structure of the paragraph not clear

·       Rows 329,330,333,359: what do these numbers stand for?

·       Row 349: addition, its production remained unaltered

·       Row 399: Bacteroides fragilis

Author Response

We would like to thank you for providing helpful comments on our manuscript. We have attempted to address all concerns, as itemized below: 

Suggestions:

Rows 84-85: how were animals grouped?

AU: Accepted, it has been revised as “Forty-four multiparous Chinese Holstein cows (2.8 ± 0.19 of parity, 772 ± 12.3 kg of body weight [BW], 65.8 ± 8.6 days in milk [DIM] and 35.3 ± 1.9 kg/d of milk production, mean ± SD) were blocked by parity, BW, DIM, and milk production, and they were allocated four treatments in a 2×2 factorial block design, three replicates per group, and 3, 4 and 4 cows for one of three replicates, respectively. ”.

Row 97: when was diet administered?

AU: Accepted, it has been revised as “fed the same standard diet ad libitum at 08:00 h and 17.30 h daily”.

Row 104: why were animals weighed at 16 h?

AU: This time point is after milking and before feeding.

Rows 111,112,121,129,134,135: how were time sampling and days intervals established?

AU: The sampling time of rumen fluid was mainly considered between the middle and end of the trial and between two feedings. Chromic oxide powder was administered backwards from the end of the trial and before each feedings. Blood samples were collected after feeding on the end of the trial. The collection time of mammary tissues was determined before the second feeding on the end of the trial.

Minor revision:

Row 18: elevated milk

AU: Accepted, it has been revised.

Row 81: from May

AU: Accepted, it has been revised.

Row 94: What does NRC stand for?

AU: Accepted, it has been revised as “nutrient requirements of dairy cattle”.

Table 2: description on the same page of the table, please check the rest of the article

AU: Accepted, it has been revised according to your suggestion.

Row 211: μ represents

AU: Accepted, it has been revised.

Table 3 in one page, please check the rest of the article, description is not clear, put a superscript per abbreviation

AU: Accepted, it has been revised according to your suggestion.

Row 244: UFA stands for?

AU: Accepted, it has been revised as “unsaturated fatty acid (UFA)”.

Table 4: saturated fatty acid (SFA), description is not clear, put a superscript per abbreviation

AU: Accepted, it has been revised according to your suggestion.

Table 5: unsaturated fatty acid (UFA), description is not clear, put a superscript per abbreviation, please check the rest of the article

AU: Accepted, it has been revised according to your suggestion.

Row 264: what does CP stand for? Every abbreviation has to be explained

AU: Accepted, it has been revised as “crude protein (CP)”.

Row 320: structure of the paragraph not clear

AU: Accepted, it has been revised.

Rows 329,330,333,359: what do these numbers stand for?

AU: Accepted, it stand for references and has been revised.

Row 349: addition, its production remained unaltered

AU: Accepted, it has been revised according to your suggestion.

Row 399: Bacteroides fragilis

AU: Accepted, it has been revised according to your suggestion.

Reviewer 5 Report

The idea of the manuscript is novel and worth trying. The manuscript is well written, the introduction, materials and methods, results and the discussion sections are well presented. However, the abstract needs some modifications to reflect the quality of the manuscript.

More specifically: 

Lines 8 - 13: as per the style of journal, please remove the abbreviations. 

Line 17: please insert more information about the materials and methods before starting with the results. It is necessary to provide the readers with the name of the treatments. The abstract must give the readers full idea about the study.

Line 19: where are the results of the digestibility?

Lines 29-31: please rewrite the summary statement to reflect the dats presented in the abstract.

Lines 80-87: some of these information must be provided in the abstract

The rest of the manuscript is well written . 

Author Response

We would like to thank you for providing helpful comments on our manuscript. We have attempted to address all concerns, as itemized below:

The idea of the manuscript is novel and worth trying. The manuscript is well written, the introduction, materials and methods, results and the discussion sections are well presented. However, the abstract needs some modifications to reflect the quality of the manuscript.

AU: Accepted, the abstract has been revised.

More specifically: 

Lines 8 - 13: as per the style of journal, please remove the abbreviations. 

AU: Accepted, the abbreviations have been removed.

Line 17: please insert more information about the materials and methods before starting with the results. It is necessary to provide the readers with the name of the treatments. The abstract must give the readers full idea about the study.

AU: Accepted, “Forty-four multiparous Chinese Holstein cows (2.8 ± 0.19 of parity, 772 ± 12.3 kg of body weight [BW], 65.8 ± 8.6 days in milk [DIM] and 35.3 ± 1.9 kg/d of milk production, mean ± SD) were grouped by parity, BW, DIM, and milk production, and they were allocated four treatments in a 2×2 factorial block design. Additional CCP (0 mg/kg [CCP-] or 55 mg/kg dry matter [DM] of calcium pantothenate from CCP [CCP+]) and CT (0 g/kg [CT-] or 5.3 mg/kg DM of thiamine from CT [CT+]) were mixed into the total mixed ration”. has been added in Line 19.

Line 19: where are the results of the digestibility?

AU: Accepted, the results of the digestibility was listed in Table 6.

Lines 29-31: please rewrite the summary statement to reflect the dats presented in the abstract.

AU: Accepted, it has been revised as “The combination of CCP and CT more effectively increased the ruminal total VFA concentation, the acetate to propionate ratio and blood glucose level, decreased ammoniacal nitrogen concentration than that achieved by CCP or CT alone. ”.

Lines 80-87: some of these information must be provided in the abstract

AU: Accepted, it has been provided in the abstract.